# Characterization of a Potent, Selective, and Safe Inhibitor, Ac15(Az8)_2_, in Reversing Multidrug Resistance Mediated by Breast Cancer Resistance Protein (BCRP/ABCG2)

**DOI:** 10.3390/ijms232113261

**Published:** 2022-10-31

**Authors:** Tsz Cheung Chong, Iris L. K. Wong, Jiahua Cui, Man Chun Law, Xuezhen Zhu, Xuesen Hu, Jason W. Y. Kan, Clare S. W. Yan, Tak Hang Chan, Larry M. C. Chow

**Affiliations:** 1Department of Applied Biology and Chemical Technology and State Key Laboratory of Chemical Biology and Drug Discovery, Hong Kong Polytechnic University, Hong Kong, China; 2School of Chemistry and Chemical Engineering, Shanghai Jiao Tong University, Shanghai 200240, China; 3Department of Chemistry, McGill University, Montreal, QC H3A 2K6, Canada

**Keywords:** multidrug resistance, breast cancer resistance protein, BCRP, ABCG2, flavonoid dimers

## Abstract

Overexpression of breast cancer resistance transporter (BCRP/ABCG2) in cancers has been explained for the failure of chemotherapy in clinic. Inhibition of the transport activity of BCRP during chemotherapy should reverse multidrug resistance. In this study, a triazole-bridged flavonoid dimer **Ac15(Az8)_2_** was identified as a potent, nontoxic, and selective BCRP inhibitor. Using BCRP-overexpressing cell lines, its EC_50_ for reversing BCRP-mediated topotecan resistance was 3 nM in MCF7/MX100 and 72 nM in S1M180 in vitro. Mechanistic studies revealed that **Ac15(Az8)_2_** restored intracellular drug accumulation by inhibiting BCRP-ATPase activity and drug efflux. It did not down-regulate the cell surface BCRP level to enhance drug retention. It was not a transport substrate of BCRP and showed a non-competitive relationship with DOX in binding to BCRP. A pharmacokinetic study revealed that I.P. administration of 45 mg/kg of **Ac15(Az8)_2_** resulted in plasma concentration above its EC_50_ (72 nM) for longer than 24 h. It increased the AUC of topotecan by 2-fold. In an in vivo model of BCRP-overexpressing S1M180 xenograft in Balb/c nude mice, it significantly reversed BCRP-mediated topotecan resistance and inhibited tumor growth by 40% with no serious body weight loss or death incidence. Moreover, it also increased the topotecan level in the S1M180 xenograft by 2-fold. Our results suggest that **Ac15(Az8)_2_** is a promising candidate for further investigation into combination therapy for treating BCRP-overexpressing cancers.

## 1. Introduction

Overexpression of the membrane-bound ATP binding cassette (ABC) transporter superfamily is one of the major mechanisms for mediating multidrug resistance (MDR) in various cancer cells. The three major ABC transporter proteins are P-glycoprotein (P-gp; ABCB1), multidrug-resistance-associated protein 1 (MRP1; ABCC1), and breast cancer resistance protein (BCRP; ABCG2) [1,2,3,4]. Structurally, P-gp and MRP1 are typical ABC transporters with two (or three) transmembrane domains (TMDs) and two highly conserved nucleotide-binding domains (NDBs). BCRP, however, is a 72-kDa half transporter [5] with only one NBD and one TMD [6]. In a recent multi-platform approach analysis of over 50,000 cancer patients [7], the expression of the ABC transporters was examined. Across all tumors profiled (*n* = 51,939), MRP1 positivity was highest at 81% (19,935/24,682), BCRP at 66% (8849/13,409), and P-gp the lowest at 23% (11,969/51,313). About 29% of the patients tested exhibited co-expression of all three transporters. The concept of co-administration of a potent inhibitor of ABC transporter with an anticancer drug to overcome MDR has been evaluated in several clinical trials but led to a disappointing outcome [8,9,10,11]. The clinical failure is due to the fact that none of the clinical trials had patient selection based on prospective evaluation of the expression of the drug transporters. For example, in the trials that evaluate P-gp inhibitors in non-small-cell lung cancer, transporters other than P-gp, such as MRP1 or BCRP, may have accounted for the drug resistance [12]. It was not surprising that inhibitors that targeted P-gp would fail to overcome MDR due to MRP1 or BCRP. Further improvement of inhibitors of ABC transporters is necessary and should focus on potency, specificity, and safety. In view of the high percentage of positivity for MRP1 (81%) and BCRP (66%) in the study cited above [7], priority should be given to the discovery of potent, selective, and safe inhibitors of MRP1 and BCRP.

In various cancers, BCRP expression highly correlates with a poor prognosis and lower survival rate of cancer patients [13,14,15]. BCRP has a broad substrate specificity including clinically used anticancer drugs such as methotrexate [16], mitoxantrone [17], topotecan [18], doxorubicin (DOX) [19], SN38 [20], and sorafenib [21]. Since its discovery in 1998 [22,23,24], extensive effort has been made to discover potent BCRP inhibitors. Fumitremorgin C (FTC), a compound isolated from *A. fumigatus*, was found to be a potent and specific BCRP inhibitor, but it causes serious neurotoxicity [25]. To reduce its side effects, several synthetic analogs of FTC have been developed, and among them, Ko143 was found to be more potent and less toxic than FTC [26]. Recently, a scaffold combination of a tetrahydroisoquinoline-ethylphenyl substructure together with anthranilic acid derivatives [27] as well as novel derivatives of acrylonitrile [28] have been reported to be potent inhibitors of BCRP. In addition, various 2,4,6-substituted quinazolines [29], 2,4-disubstituted pyridopyrimidines [30], and 4-anilino-2-pyridylquinazolines and -pyrimidines [31] have also been reported to be highly potent and nontoxic inhibitors of BCRP. However, none of these recently discovered BCRP inhibitors has yet successfully reached the clinical stage.

Many natural flavonoids abundantly present in fruits and vegetables have been found to exhibit moderate activity in inhibiting ABC transporters [32,33]. In view of the *pseudo*-dimeric structure of ABC transporters, we reasoned that coupling two flavonoid moieties together with a biocompatible linker would help to improve their transporter inhibitory activity. Indeed, we found that flavonoid dimers of general structure I (Figure 1) with polyethylene glycol (PEG) linkers showed promising P-gp- and MRP1-modulating activities with nanomolar EC_50_ values (70 to 170 nM) [34,35,36,37,38,39]. Because of their highly hydrophobic nature and unacceptably poor aqueous solubility, compounds of general structure I are not suitable drug candidates. We have since developed a “click chemistry” approach [40] to rapidly synthesize numerous triazole-bridged homo- and hetero-flavonoid dimers. Several of these flavonoid dimers were successfully identified as safe, potent MRP1 [41], and/or BCRP inhibitors [42] using in vitro studies. These triazole-bridged flavonoid dimers have improved physiochemical property due to the basic property of triazole structure [43] and the better aqueous solubility of its hydrochloride salts. One of the flavonoid dimers, namely **Ac15(Az8)_2_** (Figure 1), was found to have superior selectivity towards BCRP than that of Ko143 in vitro [42]. Here, we report the characterization of this novel inhibitor of BCRP, **Ac15(Az8)_2,_** which appears to be highly potent, selective for BCRP over P-gp or MRP1, and is effective in an in vivo model of topotecan-resistant human colon cancer xenograft in Balb/c nude mice. Furthermore, the inhibitor, in combination with the anticancer drug topotecan, appears to be safe for the animals with no serious body weight loss or death incidence.

## 2. Results

### 2.1. **Ac15(Az8)_2_** Modulates BCRP-Mediated MDR In Vitro

In our previous study, **Ac15(Az8)_2_** was found to have high in vitro BCRP-inhibitory potency in restoring topotecan cytotoxicity [42]. Here, **Ac15(Az8)_2_** was further characterized in vitro and in vivo. Human colon carcinoma cell lines S1 and S1M180 were used: S1 was the sensitive parental cell line, and S1M180 was the resistant cell line obtained by mitoxantrone selection in which BCRP was found to be overexpressed. S1M180 was resistant to topotecan and DOX by 27.9-fold and 133.7-fold when compared to its parental cell line S1, respectively (Table 1). At 0.5 μM, **Ac15(Az8)_2_** markedly reversed BCRP-mediated topotecan resistance and DOX resistance in S1M180 cells with RF of 44.6 and 40.3, respectively (Table 1).

To study the potency in more detail, we have determined the effective concentration (EC_50_) of **Ac15(Az8)_2_** and Ko143 in reversing BCRP-, P-gp-, and MRP1-mediated drug resistance. EC_50_ is defined as the concentration at which the modulator can reduce the IC_50_ of a cell line towards an anticancer by half. **Ac15(Az8)_2_** was more potent than Ko143 in reversing topotecan resistance in HEK293/R2 (EC_50_ = 5.3 nM) and MCF7-MX100 (EC_50_ = 3.3 nM) but not in S1M180 (EC_50_ = 56–72 nM) (Table 2). Both **Ac15(Az8)_2_** and Ko143 were more selective against BCRP than MRP1 and P-gp as EC_50_ values of **Ac15(Az8)_2_** and Ko143 towards BCRP were 6- to 224-fold and 42- to 909-fold lower than that of MRP1 and P-gp, respectively (Table 2). Importantly, **Ac15(Az8)_2_** was less toxic than Ko143 towards L929, a normal mouse fibroblast cell line. **Ac15(Az8)_2_** caused no toxicity towards L929 (IC_50_ > 500 µM), whereas Ko143 resulted in moderate toxicity with IC_50_ of 29 µM (Table 2).

### 2.2. **Ac15(Az8)_2_** Increases Retention of BCRP-Substrates by Inhibiting Efflux

The above results showed that **Ac15(Az8)_2_** is a potent and selective BCRP modulator as well as nontoxic to normal cells. We determined whether the modulation of BCRP-mediated topotecan and DOX resistance is associated with a concomitant increase in drug accumulation. BCRP-overexpressing S1M180 accumulated about 2.1-fold (*p* < 0.01) and 4.6-fold (*p* < 0.001) less topotecan (Figure 2A) and DOX (Figure 2B) than its parental cell line S1, respectively. **Ac15(Az8)_2_** can increase the accumulation of topotecan or DOX in S1M180 cells in a dose-dependent manner. In contrast, **Ac15(Az8)_2_** did not affect the topotecan or DOX accumulation in non-BCRP overexpressing S1 cells (Figure 2A,B). We then investigated whether **Ac15(Az8)_2_** can block BCRP-mediated drug efflux in S1M180 cells. S1M180 cells were pre-loaded with DOX and then transferred to DOX-free media to allow DOX efflux. Without **Ac15(Az8)_2_**, intracellular DOX level in S1M180 cells decreased by 50% in 30 min (Figure 2C). In the presence of 1 μM of **Ac15(Az8)_2_**, DOX efflux was significantly reduced. The intracellular DOX levels remained at 75% after 30 min (Figure 2C). The above result demonstrated that the reversal of DOX resistance by **Ac15(Az8)_2_** in S1M180 cells was due to the blockage of drug efflux, leading to an increased drug accumulation and thus restoring the chemosensitivity of the BCRP-overexpressing cell line.

BCRP is an efflux transporter with broad substrate selectivity. In order to study whether **Ac15(Az8)_2_** is itself a substrate of BCRP, its intracellular level was measured after treating the S1 or S1M180 cells for 2 h. Intracellular levels of **Ac15(Az8)_2_** in S1M180 were about 20% higher than that in S1 (Figure 2D), suggesting that **Ac15(Az8)_2_** is not a transport substrate of BCRP.

The BCRP inhibitory effect of **Ac15(Az8)_2_** was further demonstrated using confocal microscopy (Figure 3). BCRP was localized at the plasma membrane of S1M180 only but not in wild-type S1. DOX (green color) was found in S1 (DMSO control) but not in S1M180 (DMSO control). Treatment with **Ac15(Az8)_2_** in S1M180 restored the DOX level to that of S1 (Figure 3).

### 2.3. **Ac15(Az8)_2_** Does Not Affect the Cell Surface BCRP Level

Flow cytometry study indicated that cell surface BCRP was found in HEK293/R2 but not in HEK293/pcDNA3.1 cells (Figure 4A). Quantitatively, there was about 32–36-fold more cell surface BCRP in HEK293/R2 cells compared to that of HEK293/pcDNA3.1 (Figure 4B). Treatment with **Ac15(Az8)_2_** for 24 or 72 h slightly increased cell surface BCRP level in HEK293/R2 by 10–20% (Figure 4B). Together, this result suggested that **Ac15(Az8)_2_** did not significantly affect cell surface BCRP level.

### 2.4. Lineweaver–Burk and Dixon Plots Suggest a Non-Competitive Relationship between DOX and **Ac15(Az8)_2_** in Binding to BCRP

We investigated the biochemical mechanism by which **Ac15(Az8)_2_** inhibits BCRP. S1M180 cells were incubated with DOX and **Ac15(Az8)_2_** at different concentrations. DOX accumulation was measured and analyzed by Lineweaver–Burk plot. It was found that **Ac15(Az8)_2_** did not affect K_m_ but reduced V_max_ (Figure 5A). This result suggested that **Ac15(Az8)_2_** has a non-competitive inhibition relationship with DOX in binding to BCRP. Moreover, it is implied that DOX and **Ac15(Az8)_2_** are not binding to the same site on BCRP. The apparent inhibition constant (K_i_) can be calculated by either the Lineweaver–Burk plot (inset of Figure 5A) or the Dixon plot (Figure 5B). K_i_ was determined to be 230–295 nM (Figure 5A,B) which is close to the EC_50_ (56 nM) in the cell proliferation assay (Table 2). This result suggested that **Ac15(Az8)_2_** reverses BCRP-mediated DOX resistance by inhibiting DOX efflux non-competitively.

### 2.5. **Ac15(Az8)_2_** Inhibits BCRP-ATPase Activity

**Ac15(Az8)_2_** was not a transport substrate of BCRP (Figure 2D) and worked non-competitively to inhibit the BCRP-mediated DOX efflux (Figure 5). In order to study whether **Ac15(Az8)_2_** can inhibit ATPase activity of BCRP and then block transporter efflux, microsome fraction from S1M180 cells was purified, and the effect of **Ac15(Az8)_2_** on vanadate-sensitive BCRP-ATPase activity was investigated. As same as Ko143, **Ac15(Az8)_2_** inhibited vanadate-sensitive BCRP-ATPase activity in a dose-dependent manner (Figure 6). The IC_50_ was 7.5 nM and 6.5 nM for **Ac15(Az8)_2_** and Ko143, respectively (Figure 6). Maximum inhibition of BCRP-ATPase activity was achieved at around 30 nM of **Ac15(Az8)_2_** and 40 nM of Ko143, respectively. These results suggest that **Ac15(Az8)_2_** can modulate BCRP-mediated DOX efflux by inhibiting its ATPase activity.

### 2.6. Pharmacokinetics (PK) Study of **Ac15(Az8)_2_** and Its Effect on PK of Topotecan in Mice

Intraperitoneal (I.P.) injection of 45 mg/kg **Ac15(Az8)_2_** resulted an AUC_(0-inf)_ of 83.5 μg·h/mL. T_max_ was at 3 h, and C_max_ was 15.7 μg/mL. Half-life (E-phase) was 10.6 h (Figure 7A). I.P. injection of 45 mg/kg **Ac15(Az8)_2_** can maintain its plasma concentration above its EC_50_ (72 nM for reversing topotecan resistance in S1M180) for longer than 24 h (Figure 7A).

BCRP is mainly found in the liver, kidney, intestine, placenta, mammary gland, and blood–brain barrier [44,45]. It affects the pharmacological and toxicological behavior of many of its substrates, such as drugs or toxins. Here, the effect of **Ac15(Az8)_2_** on topotecan PK was examined. I.P. injection of 45 mg/kg **Ac15(Az8)_2_** increased AUC of topotecan by about 2-fold, from 4.4 μg·h/mL to 8.7 μg·h/mL (Figure 7B). The half-life was prolonged from 1.3 h to 1.6 h (Figure 7B). The above results suggested that co-administration of **Ac15(Az8)_2_** does affect the PK of topotecan in plasma, but not significantly. 

### 2.7. **Ac15(Az8)_2_** Reverses BCRP-Mediated Topotecan Resistance In Vivo by Increasing the Topotecan Concentration in Tumor

Here, we established an in vivo topotecan-resistant human colon xenograft model in female Balb/c nu nu mice. Without any treatment, parental S1 xenograft volume increased by 700% in 11 days (Figure 8A). Treatment with either topotecan alone (2 mg/kg; I.P. every other day starting from day 1 for four times) or in combination with 45 mg/kg of **Ac15(Az8)_2_** (I.P. every other day starting from day 1 for four times) can significantly reduce xenograft volume (Figure 8A). Xenograft grew to 200% on the 11th day. This result suggested that S1 xenograft was highly sensitive to topotecan treatment. In contrast, the S1M180 xenograft was not responsive to four rounds of topotecan treatment (Figure 8B). Such topotecan resistance was likely due to the expression of BCRP in S1M180 xenograft. We tested if **Ac15(Az8)_2_** can reverse such BCRP-mediated topotecan resistance. On day 55, all animals receiving solvent control had been sacrificed due to a large tumor burden. Combination treatment resulted in about 40% reduction (*p* < 0.01) in tumor volume when compared with the topotecan alone group on days 55 and 71 post treatment (Figure 8B). Combination treatment also reduced tumor weight by 1.6-fold (*p* < 0.01) and 1.9-fold (*p* < 0.001) when compared with topotecan alone and the solvent groups, respectively (Figure 8C). In addition, combination treatment lengthened the tumor doubling time by 1.8 days and 4.6 days when compared with topotecan alone and solvent groups, respectively (Figure 8D). No animal death was reported during the in vivo efficacy study (Figure 8D and Appendix A). In both topotecan and combination treatments, the mice did not suffer a significant loss in body weight, and only from −4% to 18% of weight change was noted (Figure 8E). Topotecan alone at 2 mg/kg I.P. resulted in some toxicity to the mice as the body weight was lower than that of solvent control (Figure 8E). On the other hand, **Ac15(Az8)_2_** did not potentiate the toxicity of topotecan because a further decrease in body weight was not observed. These results suggested that I.P. administration of **Ac15(Az8)_2_** (45 mg/kg) was effective in modulating BCRP-mediated topotecan resistance in the in vivo S1M180 xenografted Balb/c nude mice model without inducing any mice death or significant weight loss.

We studied if **Ac15(Az8)_2_** can suppress S1M180 xenograft growth by inhibiting BCRP transport activity and increasing the intra-tumoral drug concentration. We found that the plasma level of topotecan at 5 h post administration (I.P.; 6 mg/kg) did not change significantly after co-treatment with **Ac15(Az8)_2_** (Figure 8F). In contrast, intratumor level of topotecan (I.P.; 6 mg/kg) was increased by 2-fold (*p* < 0.01) after co-treatment with **Ac15(Az8)_2_** (Figure 8F). This result is somewhat different from the pharmacokinetic study reported in Figure 7B, possibly due to the different dosages of topotecan used in the two experiments or whether xenograft was present in the mice or not. The result suggests that **Ac15(Az8)_2_** can be delivered to the xenograft, inhibited the transport activity of BCRP, enhanced the intra-tumoral topotecan retention, and finally suppressed the tumor growth.

## 3. Discussion

Since the discovery of ABC transporters and their roles in mediating MDR in cancer chemotherapy, it has been thought that the co-administration of ABC transporter inhibitor together with an anticancer drug can reverse drug resistance. Unfortunately, several clinical trials using this approach to overcome MDR have led to disappointing outcomes [11]. Some factors which have been suggested for the lack of success in the inhibitors are: (1) enhanced toxicity at the non-target site, (2) drug–drug interaction, namely, the effect of the inhibitor on the PK of the anticancer drug [46], and (3) adversely modify drug distribution in solid tumors [46]. Research on new inhibitors should therefore focus on these factors by discovering compounds of high potency, high selectivity with low toxicity, and improved PK to minimize drug-drug interaction.

The triazole-bridged flavonoid dimer **Ac15(Az8)_2_** has been found to be a potent, nontoxic, and BCRP-selective inhibitor in vitro (Table 2). It can reverse topotecan and DOX resistance in three BCRP-overexpressed cell lines with EC_50_ ranging from 3 nM to 72 nM (Table 2). More importantly, **Ac15(Az8)_2_** was not toxic to L929 mouse fibroblast in vitro (IC_50_ > 500 µM), whereas Ko143 displayed moderate toxicity (IC_50_ = 29.2 µM) (Table 2). The safety, PK, and efficacy of **Ac15(Az8)_2_** were evaluated in animal studies. **Ac15(Az8)_2_** exhibited promising plasma bioavailability after I.P. injection (Figure 7A). Administration of 45 mg/kg of **Ac15(Az8)_2_** was enough to maintain a plasma level of **Ac15(Az8)_2_** above its in vitro EC_50_ (72 nM for reversing topotecan resistance in S1M180) for more than 24 h (Figure 7A). **Ac15(Az8)_2_** (45 mg/kg) exhibited potent BCRP inhibitory activity and restored the anti-tumor activity of topotecan (2 mg/kg) in S1M180 xenograft. **Ac15(Az8)_2_** at this dosage was able to enhance the intra-tumoral topotecan concentration by nearly two folds (Figure 8F) and did not adversely modify drug distribution in the solid tumor. Regarding possible drug-drug interaction between the inhibitor and topotecan, co-administration of **Ac15(Az8)_2_** did affect the PK of topotecan in plasma, but not significantly (Figure 7B). This is consistent with the observation that **Ac15(Az8)_2_** (45 mg/kg) together with topotecan (2 mg/kg) did not lead to significant body weight reduction or animal death when compared with topotecan alone (Figure 8D,E).

From the mechanistic studies, intracellular topotecan and DOX level of S1M180 cells can be completely restored to a wild-type level by **Ac15(Az8)_2_** in a dose-dependent manner (Figure 2A,B). Modulation of BCRP-mediated resistance by **Ac15(Az8)_2_** is not due to the alteration of surface BCRP protein level (Figure 4) but due to the inhibition of BCRP-ATPase activity and blockage of BCRP-mediated efflux (Figure 2C and Figure 6). **Ac15(Az8)_2_** was shown to inhibit BCRP-ATPase activity with an EC_50_ of 7.5 nM, which was comparable to that of Ko143 (6.5 nM) (Figure 6). **Ac15(Az8)_2_** was not a transport substrate of BCRP (Figure 2D) and showed a non-competitive relationship with DOX for binding to BCRP (Figure 5), suggesting that **Ac15(Az8)_2_** and DOX do not compete for the same binding site on BCRP. In silico molecular docking studies, apigenin was found to bind the NBDs of P-gp [47]. The NBD is a highly conserved domain among ABC transporters. It is possible that **Ac15(Az8)_2_** might bind to the NBDs of BCRP and then inhibit the ATPase activity. Its binding site on BCRP is under active investigation using a photoaffinity crosslinking approach. 

## 4. Materials and Methods

### 4.1. Reagent and Materials

Dimethyl sulfoxide (DMSO), N-methyl-2-pyrrolidone (NMP), doxorubicin (DOX), Cremophor-EL, and Ko143 were purchased from Sigma-Aldrich. Topotecan was purchased from Shanghai Aladdin Bi-Chem Technology Co., Ltd., Shanghai, China. Human breast cancer cell line MCF7 was purchased from ATCC. MCF7/MX100, human embryonic kidney cell lines HEK293/pcDNA3.1, BCRP-transfectant HEK293/R2, human colon carcinoma cell lines S1 and S1M180 were kindly provided by Dr. S. E. Bates (Columbia University/New York Presbyterian Hospital, Manhattan, NY, USA). The human breast cancer cell line, LCC6MDR, was kindly provided by Dr. Robert Clarke (Georgetown University, Washington, DC, USA). Ovarian cancer cell line, 2008/MRP1, was a generous gift of Dr. Piet Borst (The Netherlands Cancer Institute, Amsterdam, The Netherlands). The mouse fibroblast cell line, L929, was purchased from ATCC. 

### 4.2. Chemical Synthesis of **Ac15(Az8)_2_**

Compound **Ac15(Az8)_2_** was prepared according to previous report [42] with 99% purity (Appendix A). Conversion to its hydrochloride salt was accomplished by adding concentrated hydrochloric acid followed by evaporation to dryness. A 10 mM stock solution of the free base form of **Ac15(Az8)_2_** was prepared by dissolving in DMSO for in vitro studies. A stock solution of **Ac15(Az8)_2_** for in vivo experiment was prepared by dissolving the hydrochloride salt of **Ac15(Az8)_2_** in NMP, followed by the addition of Cremophor-EL and then diluted to 4 mg/mL by H_2_O (ratio of NMP, Cremophor-EL and water 1:1:8).

### 4.3. Cell Culture

MCF7, MCF7-MX100, HEK293/pcDNA3.1, HEK293/R2, S1, and S1M180 were cultured in RPMI 1640 medium (Gibco BRL) with 10% FBS (Hyclone) and 100 U/mL penicillin and 100 μg/mL of streptomycin (Gibco BRL) and maintained at 37 °C in a humidified atmosphere with 5% CO_2_. The cells were split constantly after the formation of a confluent monolayer. To split the cells, the plate was washed briefly with phosphate-buffered saline (PBS), treated with 0.05% trypsin-EDTA (Gibco BRL), and harvested by centrifugation.

### 4.4. Cell Proliferation Assay

In total, 6500 cells of S1M180 or MCF7-MX100 were incubated with various doses of topotecan or DOX and modulators. The final volume in each well of 96-well plates was 200 µL. The plates were then incubated for 5 days at 37 °C. The CellTiter 96 AQ_ueous_ Assay (Promega) was used to measure cell proliferation [42]. According to the manufacturer’s instructions, MTS (2 mg/mL) and PMS (0.92 mg/mL) were mixed in a ratio of 20:1. An aliquot (10 μL) of the freshly prepared MTS/PMS mixture was added into each well, and the plate was incubated for 2 h at 37 °C. Optical absorbance at 490 nm was then recorded with microplate absorbance reader (Bio-Rad). IC_50_ values were calculated from the dose-response curves of MTS assays (Prism 4.0).

### 4.5. Drug Accumulation Assay

Briefly, 1 × 10^6^ cells of S1 and S1M180 cells were pre-incubated with 100 nM or 1000 nM **Ac15(Az8)_2_** for 2 h and followed by 50 µM of topotecan or 20 µM of DOX for 1-h incubation. DMSO at 0.1% was used as a negative control. After incubation, the cells were spun down and washed with cold PBS, pH7.4 for 2 times and lysed with 100 μL of lysis buffer (0.75 M HCl, 0.2% Triton-X100 in isopropanol) with vigorous vortex. The lysate was further incubated at 37 °C for 20 min. After incubation, the lysate was spun down, and 100 μL of supernatant was saved and seeded into a black 96-well microtiter plate. The fluorescence level of DOX was determined by a fluorescence microplate reader (BMG Technologies) using excitation and an emission wavelength pair of 460 nm and 610 nm [38].

### 4.6. Immunofluorescence Staining

For DOX level determination, 1 × 10^6^ cells were seeded on a sterilized glass coverslip in a 24-well plate overnight at 37 °C with 5% CO_2_, and 1 µM of DOX was added to the cells in the presence or absence of 1 µM **Ac15(Az8)_2_**. After overnight incubation, cells were washed with PBS and fixed with 4% paraformaldehyde. The cells were washed and blocked with a blocking solution (3% bovine serum albumin and 0.1% triton X-100 in PBS). The cells were incubated with primary antibody (BCRP antibody BXP21 (Santa Cruz), diluted at 1:1000) and secondary antibody (donkey anti-mouse IgG(H + L) conjugated with Alexa Fluor 594 (Invitrogen), diluted at 1:500 (excitation: 561 nm; emission: 617 nm). The excitation wavelength for DOX was 488 nm, and emission wavelength was 590 nm.

### 4.7. Determination of Surface BCRP Protein Expression

Forty thousand HEK293/pcDNA3.1 and HEK293/R2 cells were seeded in a 6-well plate and incubated with 0, 1, or 3 µM of **Ac15(Az8)_2_** for 24 h or 72 h, respectively. After incubation, the cells were trypsinized and washed once with 1X PBS. The cells were resuspended in 50 µL FACS buffer (1% BSA and 1 mM EDTA in PBS) and stained with 2.5 µL FITC mouse anti-human BCRP antibody (Miltenyi Biotec) at 4 °C for 45 min. After staining, the cells were washed once with 500 µL cold FACS buffer and resuspended in 200 µL FACS buffer. The BCRP-FITC level was analyzed by BD C6 Accuri flow cytometer using FL1 channel at EX 480 nm and EM 533/30 nm. For each sample, 30,000 events were collected [41].

### 4.8. Vanadate-Sensitive BCRP-ATPase Activity

The membrane fraction of S1M180 cells was prepared as previously reported [48]. Around 5 × 10^7^ cells of S1M180 were resuspended in 5 mL homogenization buffer (0.33 M sucrose, 300 mM Tris pH7.4, 1 mM EDTA, 1 mM EGTA, 2 mM DTT, 100 mM 6-aminocaproic acid, 1 mM PMSF, and 1X protease inhibitor (cOmplete™ Protease Inhibitor Cocktail Tablets, Roche) and lysed using a Branson SFX550 sonicator for 10 cycles at 50% amplitude with 30 s on/30 s off. The lysate was centrifuged at 3500× *g* for 10 min at 4 °C. Membrane fraction of cells was collected by ultracentrifugation of cell lysate at 45,000 rpm using Himac CP70G (Hitachi) for 1.5 h. Membrane fraction pellet was re-suspended in 300 µL of ATPase assay buffer (50 mM Tris at pH7.5, 2 mM EGTA at pH 7.0, 2 mM DTT, 50 mM KCl, 10 mM MgCl_2_, 5 mM sodium azide, and 1 mM ouabain). Vanadate-sensitive BCRP-ATPase activity of **Ac15(Az8)_2_** or Ko143 was conducted as previously reported [48]. In brief, membrane fraction was pre-incubated with or without 0.3 mM sodium *ortho*vanadate and respective tested compounds for 30 min. Then, 2.5 mM ATP was added to each well, and the plate was further incubated for 1 h at 37 °C. Reactions were stopped by adding 200 µL freshly prepared cold stop buffer (0.2% ammonium molybdate, 1.4% sulphuric acid, 0.9% SDS, and 1% ascorbic acid) and incubated at room temperature for 15 min. The absorbance of 655 nm was measured by CLARIOstar^®^ microplate reader (BMG). Sodium orthovanadate inhibits ABC transporter ATPase activity and is used to calculate vanadate-sensitive activity by subtraction from the total ATPase activity (without sodium orthovanadate). 

### 4.9. DOX Efflux Study

To measure the DOX efflux, S1M180 cells were pre-incubated with 20 µM DOX for 1 h at 37 °C. After 1-h incubation, the cells were spun down and washed once with cold PBS. Then, the cells were further incubated with or without compound **Ac15(Az8)_2_** (1 μM) in RPM1640-supplemented media. At 0, 5, 10, 30, 60, 120, and 180 min, 1 × 10^6^ cells were harvested for measuring the intracellular DOX concentration as described in Section 4.5. The % of DOX reduction was calculated = [(DOX level at final time point/DOX level at 0 min) × 100%].

### 4.10. Ultra-Performance Liquid Chromatography–Tandem Mass Spectrometry (UPLC–MS/MS)

**Ac15(Az8)_2_** and topotecan in plasma were analyzed by UPLC–MS/MS. UPLC (Acquity Waters) and triple quadrupole mass analyzer (Quattro Ultima) with an electrospray ionization source in the positive mode were used to quantify **Ac15(Az8)_2_** and topotecan. Acquity UPLC BEH C8 and C18 column (1.7 µm 2.1 × 50 mm) from Waters was used to separate **Ac15(Az8)_2_** and topotecan, respectively. Mobile phases used for **Ac15(Az8)_2_** analysis were water and acetonitrile. **Ac15(Az8)_2_** and its internal standard **Ac15(Az5)_2_** (Appendix A) [42] were monitored at *m/z* 608 → *m/z* 149 and *m/z* 594 → *m/z* 1097, respectively. Mobile phases used for topotecan analysis were water and methanol. Topotecan and its internal standard tetracycline were monitored at *m/z* 422 → *m/z* 377 and *m/z* 445 → *m/z* 410, respectively.

### 4.11. Pharmacokinetic Studies

Female Balb/c mice at 4–6 weeks were used for the pharmacokinetic studies. **Ac15(Az8)_2_** at 4 mg/mL was prepared in 10% NMP, 10% Cremophor-EL and 80% H_2_O. Topotecan was dissolved in H_2_O at 1 mg/mL, and 45 mg/kg **Ac15(Az8)_2_** was administered through intraperitoneal (I.P.) injection. Blood was collected at 10, 30, 60, 70, 90, 120, 180, 240, 300, 480, 720 and 1440 min by cardiac puncture and stored in heparinized tubes for centrifugation at 16,000 g for 5 min. For topotecan pharmacokinetic study, 2 mg/kg topotecan was I.P. administered one hour after 45 mg/kg **Ac15(Az8)_2_** or solvent injection. Blood was collected at 10, 30, 60, 120, 240 and 480 min post-topotecan administration. All animal studies were approved by the Animal Subjects Ethics Sub-committee of the Hong Kong Polytechnic University and performed in accordance with the Cap340 Animal License of the Department of Health in Hong Kong.

### 4.12. In Vivo Efficacy of **Ac15(Az8)_2_** on Modulating Topotecan Resistance in S1M180 Xenograft

Ten million of S1M180 cells were injected subcutaneously into Balb/c nu nu mice (Beijing Vital River Laboratory Animal Technology Co., Ltd., Beijing, China). Mice with solid tumors formed were sacrificed. The tumor was removed and cut into 1 mm^3^ and xenografted subcutaneously onto another Balb/c nu nu mouse for passage or efficacy study. Treatment of nude mice started when the xenograft reached 100–150 mm^3^_._ The mice were randomized into 3 groups (*n* = 7 mice per group). Group (1) was treated with **Ac15(Az8)_2_** solvent (I.P.) and topotecan solvent (I.P.), group (2) was treated with **Ac15(Az8)_2_** solvent (I.P.) and topotecan (2 mg/kg, I.P.), group (3) was treated with **Ac15(Az8)_2_** (45 mg/kg, I.P.) and topotecan (2 mg/kg, I.P.). The formulation used was the same as that of the pharmacokinetic study. S1M180-bearing nude mice were treated for 4 rounds. In each round, they were treated once every two days 6 times, allowed to rest for 7 days, and treated again. There was a total of 24 treatments for group (2) and group (3), whereas group (1) was treated only 18 times as the animals in that control group had tumor burden and therefore needed to be sacrificed. The body weights of mice were monitored throughout the experiment. Tumor dimensions were measured by an electronic caliper, and the tumor volume was calculated as reported previously [39].

### 4.13. In Vivo Topotecan Accumulation in Tumor

In total, 45 mg/kg of **Ac15(Az8)_2_** or **Ac15(Az8)_2_** solvent was I.P. administered into S1M180-bearing mice one hour prior to 6 mg/kg of topotecan injection (I.P.). Mice were sacrificed after 5 h post-topotecan administration. The excised tumor was homogenized with 2:1 PBS: tumor (*v/w*). Homogenate was mixed with methanol in 1:1 in the presence of internal standard tetracycline. Samples were centrifuged at 16,000 g for 5 min. The supernatant was filtered through a 0.2 µm nylon filter before UPLC–MS/MS analysis.

## 5. Conclusions

In summary, the flavonoid dimer **Ac15(Az8)_2_** is a potent, safe, and selective BCRP inhibitor. It significantly reverses BCRP-mediated drug resistance in vitro and in vivo in human colon cancer xenograft experiments. **Ac15(Az8)_2_** inhibits the BCRP-ATPase activity and drug efflux restores intracellular drug accumulation, and eventually chemo-sensitizes the BCRP-overexpressing cells or xenograft to anticancer drug again. Therefore, **Ac15(Az8)_2_** appears to meet many of the criteria as a potential candidate for further investigation into combination therapy for treating BCRP-overexpressing cancer.

## Figures and Tables

**Figure 1 ijms-23-13261-f001:**
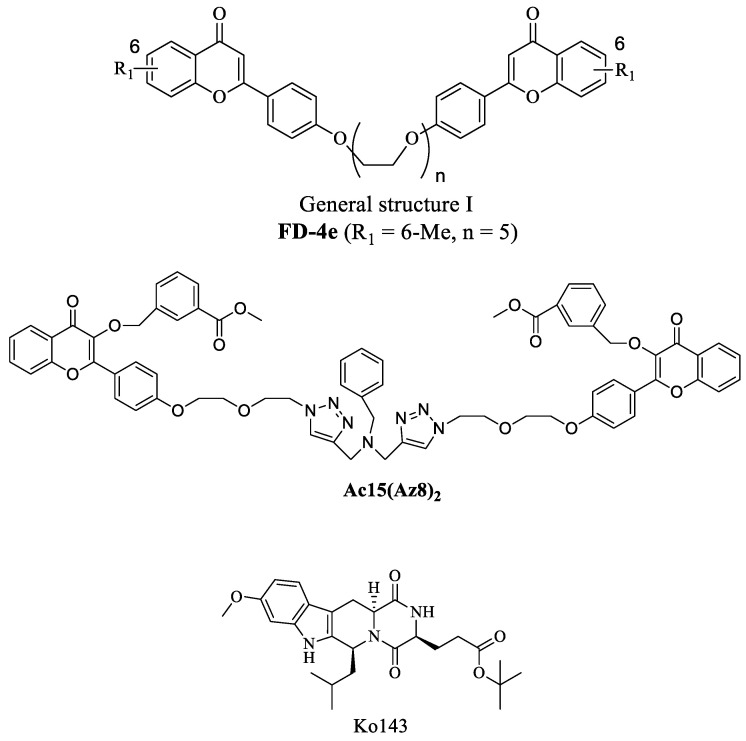
Chemical structure of general structure I, **Ac15(Az8)_2_** and Ko143.

**Figure 2 ijms-23-13261-f002:**
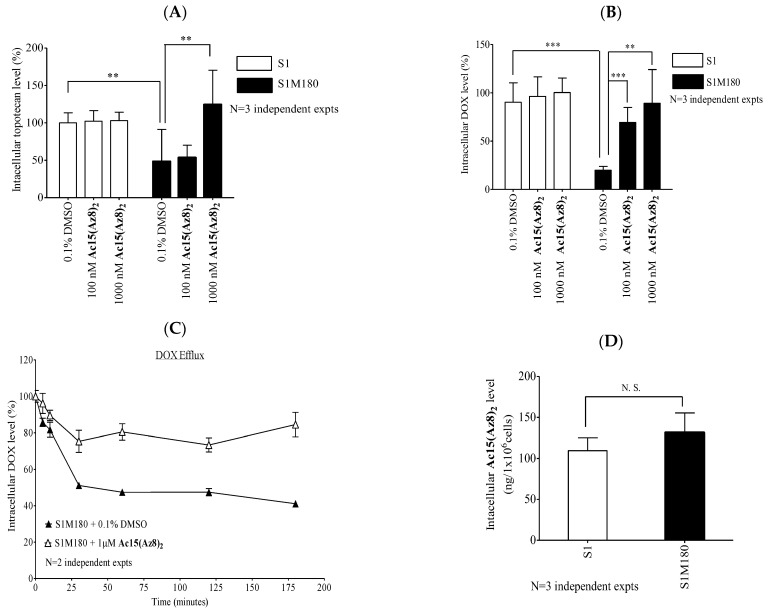
Effect of **Ac15(Az8)_2_** on intracellular accumulation of BCRP-substrates in S1 and S1M180 cells. Cells were pre-treated with 0.1% DMSO, 100 nM, or 1000 nM **Ac15(Az8)_2_** and then incubated with BCRP substrates. (**A**) Topotecan accumulation. (**B**) DOX accumulation. The intracellular level of drug after treatment was normalized to that of S1 cells after incubating with 0.1% DMSO. Student unpaired two-tailed *t*-test was conducted as indicated (** *p* < 0.01, *** *p* < 0.001). (**C**) To measure DOX efflux, S1M180 cells were pre-incubated with medium containing 20 μM DOX for 1 h at 37 °C. Cells were then washed and further incubated with or without 1 µM of **Ac15(Az8)_2_**. At 0, 5, 10, 30, 60, 120, and 180 min, the cells were harvested to measure the intracellular DOX level. (**D**) To determine intracellular **Ac15(Az8)_2_** concentration, 1 × 10^7^ cells were incubated with 1 µM of **Ac15(Az8)_2_** at 37 °C for 2 h. Cells were then washed with ice-cold PBS. To the cell pellet, acetonitrile was added to lyse the cells, and the supernatant was saved for **Ac15(Az8)_2_** determination using UPLC–MS/MS. All values in this figure were presented as mean ± S.D. N.S. = not significant.

**Figure 3 ijms-23-13261-f003:**
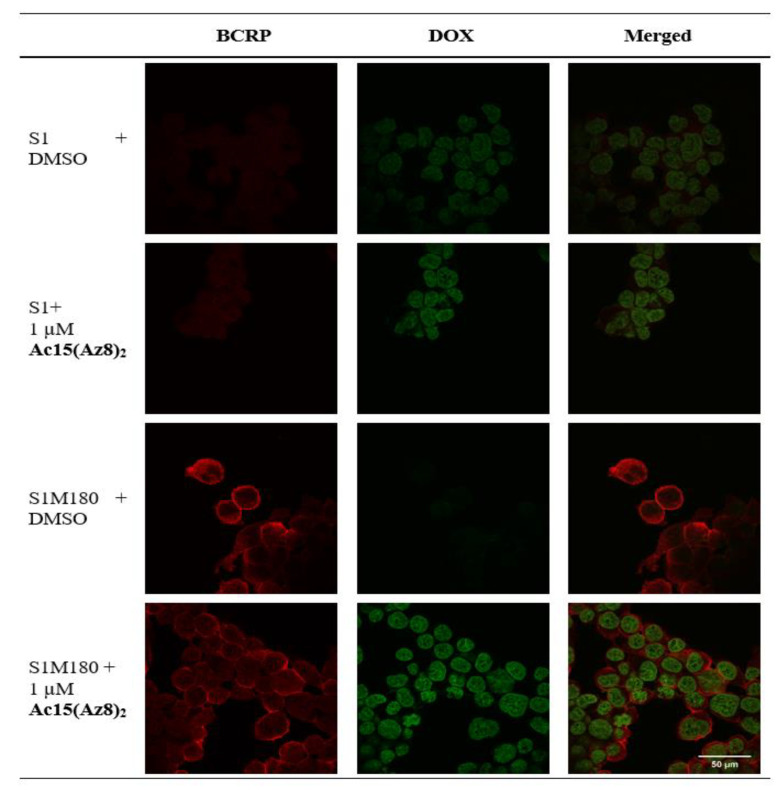
Localization of BCRP and the intracellular DOX retention in S1 and S1M180 cells were studied by confocal microscopy. After 24-h incubation with DOX in the presence of DMSO or **Ac15(Az8)_2_**. S1 or S1M180 cells were washed and stained with anti-BCRP antibody BXP-21 (shown as red). DOX fluorescence was monitored at 488 nm excitation and 590 nm for emission (shown as green).

**Figure 4 ijms-23-13261-f004:**
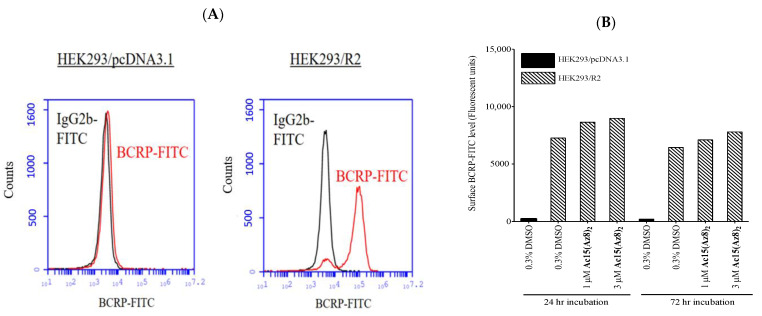
Effect of **Ac15(Az8)_2_** on surface BCRP protein expression level. (**A**) Cell surface BCRP level of HEK293/pcDNA3.1 and HEK293/R2 cells was measured by flow cytometer using BCRP-FITC antibody. Mouse IgG2b-FITC is an isotype control. (**B**) The surface BCRP level of HEK293/R2 cells was detected after incubating with 1 or 3 µM of **Ac15(Az8)_2_** for 24 h or 72 h, respectively; 0.3% of DMSO was used as solvent control.

**Figure 5 ijms-23-13261-f005:**
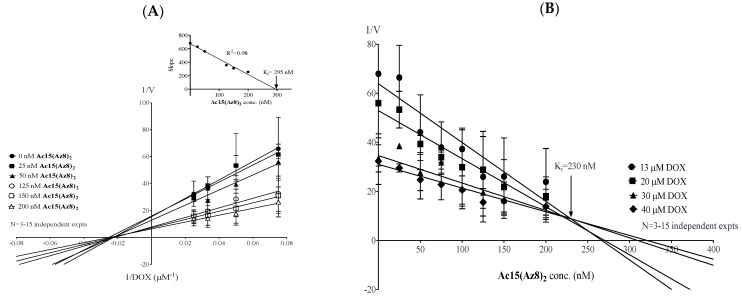
Lineweaver–Burk and Dixon plot for **Ac15(Az8)_2_** and DOX. S1M180 cells were incubated with different concentrations of **Ac15(Az8)_2_** and DOX. The cells were pre-treated with **Ac15(Az8)_2_** for 2 h and then incubated with DOX for 1 h at 37 °C. After incubation, the cells were washed with ice-cold PBS, and the intracellular DOX level was measured. (**A**) In Lineweaver–Burk plot, the reciprocal of DOX retention rate (V, μM/min/500,000 cells) is plotted against reciprocal of DOX concentration used. Six lines of **Ac15(Az8)_2_** pass through the same intercepts at the *x*-axis, which is K_m_ for the DOX. The apparent K_i_ value is determined (295 nM) by linear regression analysis from the slope of double reciprocal plots versus the concentration of **Ac15(Az8)_2_** (inset of Figure 5A). (**B**) In Dixon plot, reciprocal of DOX retention rate (1/V) was plotted against concentration of **Ac15(Az8)_2_**. Four lines of DOX intercept at a point, which is the K_i_ (230 nM) for the **Ac15(Az8)_2_**. Each data point is presented as mean ± S.D.

**Figure 6 ijms-23-13261-f006:**
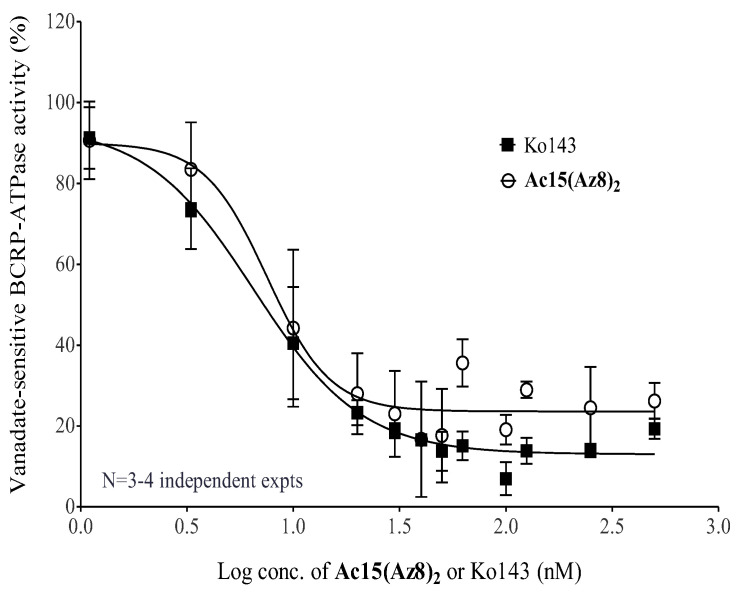
Effect of **Ac15(Az8)_2_** on vanadate-sensitive BCRP-ATPase activity. Ouabain (Na^+^/K^+^-ATPase inhibitor) and sodium azide (F-type ATPase inhibitor) were added to S1M180 microsome fraction to inhibit non-ABC transporter ATPase activities. Sodium orthovanadate inhibits ABC transporter ATPase activity and is used to calculate vanadate-ensitive activity by subtraction from the total ATPase activity (without sodium orthovanadate). S1M180 microsome fractions were pre-incubated with different doses of **Ac15(Az8)_2_** or Ko143 in the presence or absence of sodium orthovanadate at 37 °C for 30 min, followed by 1-h incubation of 2.5 mM ATP. After reaction was stopped, inorganic phosphate level was determined using colorimetry method. The percentages of vanadate-sensitive BCRP-ATPase activity were shown as mean ± S.D.

**Figure 7 ijms-23-13261-f007:**
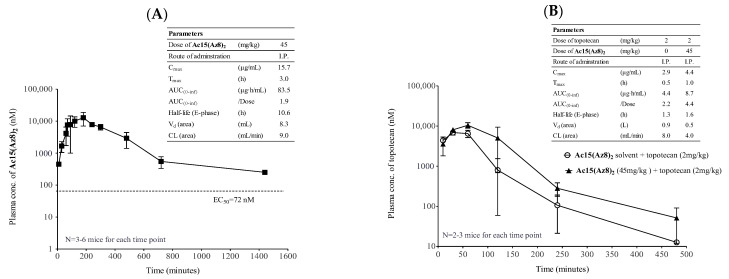
Pharmacokinetics of **Ac15(Az8)_2_** and its effect on PK of topotecan. (**A**) Plasma concentration of **Ac15(Az8)_2_** in mice was determined after I.P. injection of 45 mg/kg of **Ac15(Az8)_2_**. The dash line indicates the EC_50_ of **Ac15(Az8)_2_** (72 nM) for reversing topotecan resistance in S1M180 cell line. (**B**) Plasma concentration of topotecan in mice was determined after I.P. injection of 45 mg/kg **Ac15(Az8)_2_** or solvent one hour prior to 2 mg/kg topotecan administration (I.P.). PK solutions (PK Solutions^TM^ 2.0.3 software, Summit Research Service) was used to calculate the pharmacokinetic data.

**Figure 8 ijms-23-13261-f008:**
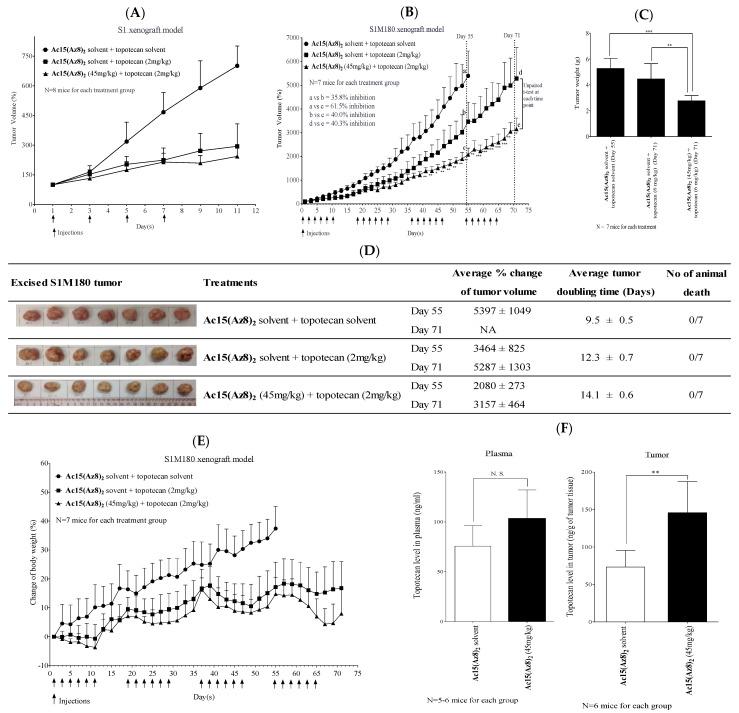
In vivo efficacy of **Ac15(Az8)_2_** in reversing BCRP-mediated topotecan resistance in S1M180 xenograft. (**A**) S1 xenograft. (**B**) S1M180 xenograft. There were 3 treatment groups in S1 and S1M180 xenograft model. Treatments were given as indicated by an arrow (↑). % of tumor volume reduction was compared among different treatment groups on days 55 and 71. a, b, and c = mean tumor volume of solvent control, topotecan alone, and combination treatment groups on day 55, respectively. d and e = mean tumor volume of topotecan alone and combination treatment groups on day 71. (**C**) Tumor weight of S1M180-bearing mice on termination day. (**D**) % change of tumor volume and tumor doubling time of S1M180 xenograft were determined on days 55 and 71. On day 55, all mice in solvent treatment group were sacrificed because of over-sized tumor. The % change in tumor volume of solvent group on day 71 was therefore labeled as NA (not applicable). Number of animal death was monitored during the efficacy study. (**E**) Body weight of S1M180-bearing mice. (**F**) To measure plasma and intratumor level of topotecan, S1M180-bearing mice were administered with 45 mg/kg (I.P.) of **Ac15(Az8)_2_** one hour prior to 6 mg/kg (I.P.) of topotecan administration. Five hours after topotecan administration, the mice were sacrificed. Plasma and tumor were collected for analyzing topotecan level using UPLC–MS/MS. Student’s unpaired two-tailed *t*-test was performed between combination group and topotecan alone group or solvent control group as indicated (* *p* < 0.05, ** *p* < 0.01, *** *p* < 0.001). All values in this figure were presented as mean ± S.D.

**Table 1 ijms-23-13261-t001:** Effect of **Ac15(Az8)_2_** on reversing topotecan and DOX resistance in S1M180 cells.

	IC_50_ (nM) of Anticancer Drugs			
Compounds	Topotecan	RF	DOX	RF
S1M180 + 0.5 μ M **Ac15(Az8)_2_**	279 ± 80	44.6	332 ± 60	40.3
S1M180 + 0.1% DMSO	12,443 ± 1186	1.0	13,367 ± 650	1.0
S1 + 0.1% DMSO	446 ± 20	27.9	100 ± 20	133.7

RF (Relative Fold) = IC_50_ of anticancer drug of a cancer cell line without **Ac15(Az8)_2_**/IC_50_ of anticancer drug of a cell line with **Ac15(Az8)_2_**. Wild type S1 cell line was incubated at 0.1% DMSO when determining the IC_50_ values.
IC_50_ values were presented as mean ± S.D. *n* = 3 independent experiments.

**Table 2 ijms-23-13261-t002:** EC_50_ of **Ac15(Az8)_2_** or Ko143 for reversing BCRP-, MRP1- and P-gp-mediated drug resistance.

Compounds	EC_50_ (nM) needed for reversing MDR						IC_50_ Towards L929 (μM)
	BCRP-Mediated Topotecan Resistance HEK293/R2	BCRP-Mediated Topotecan Resistance MCF7-MX100	BCRP-Mediated Topotecan Resistance S1M180	BCRP-Mediated DOX Resistance S1M180	MRP1-Mediated DOX Resistance 2008/MRP1	P-gp-Mediated Paclitaxel Resistance LCC6MDR	
**Ac15(Az8)_2_**	5.3 ± 2.7 *	3.3 ± 4.1 *	72.0 ± 12.0	56.1 ± 8.2	457.5 ± 10.6 *	>3000	>500
Ko143	8.7 ± 4.9 *	9.0 ± 2.1 *	9.8 ± 3.0	9.3 ± 8.3	1950.0 ± 353.6 *	1060.0 ± 207.8 *	29.2 ± 2.8 *

EC_50_
is defined as the concentration at which the modulator can reduce the IC_50_ of a cell line towards an anticancer by half. Cytotoxicity of **Ac15(Az8)_2_** and Ko143 towards normal mouse fibroblast cells L929 was determined. * The EC_50_ values for reversing drug resistance have been reported. Adapted with permission from Ref. [42]. They are included here for comparison. All values were presented as mean ± S.D. *n* = 3–4 independent experiments.

## Data Availability

The data presented in this study are available on request from the corresponding author.

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
