# Peer review of "Characterization of a Potent, Selective, and Safe Inhibitor, Ac15(Az8)2, in Reversing Multidrug Resistance Mediated by Breast Cancer Resistance Protein (BCRP/ABCG2)"

_ijms, 2022, doi:10.3390/ijms232113261_

Round 1
Reviewer 1 Report
In the present study the authors investigated inhibition of BCRP tranasporter bynovel flavonoid agent Ac15(Az8)2. The results are sound and the paper is well written. However, I have got some minor concerns dealing with the presentation of the results.
- Fig. 1. not well visible; the fragment of the image is lacking.
- Cell lines used in the experiments which results are presented in Table 2 should be also decribed in Materials and Methods section.
- Fig. 2 is too big and difficult to follow. Please divide it into 2-3 smaller figures.
- Fig. 3 – the letters A and B are missing.
- Fig. 4 – the letters A and B are missing.
- Fig. 6 is too big and difficult to follow. Please divide it into 2-3 smaller figures.
- Figures – the numbers indicating fold change are unnecessary in the images and they make them less clear.
- Materials and Methods – please provide some more details at to methodology, instead saying “as previously”. E.g., Sections 4.4.; 4.5; 4.7; 4.8
- Section 4.9. how intracellular DOX concentration was measured?
Reviewer 2 Report
None.
Reviewer 3 Report
Journal: IJMS
Title: Characterization of a Potent, Selective and Safe Inhibitor, Ac15(Az8)2, in Reversing Multidrug Resistance Mediated by Breast Cancer Resistance Protein (BCRP/ABCG2)
Authors: Tsz Cheung Chong, et al.
The authors identified a triazole-bridged flavonoid dimer Ac15(Az8)2 as a potent, safe and selective BCRP inhibitor, which reverses BCRP-meditated drug resistance in vitro and in vivo. There are a few comments and questions that should be answered by the authors.
How were the dose of Ac15(Az8)2 (45 mg/kg, I.P.) and dosing interval (every other day) determined in in vivo experiments using S1M180 xenograft? Are there any basic experiments to determine the dose and dosing interval of Ac15(Az8)2?
The authors said that “Ac15(Az8)2 was not a transport substrate of BCRP and worked non-competitively to inhibit the BCRP-mediated DOX efflux”. Does Ac15(Az8)2 bind to other transport substrate site of BCRP that DOX does not binds to? Is it possible that Ac15(Az8)2 binds to ATP binding site of BCRP? Please discuss the inhibitory mechanism of BCRP-mediated transport function by Ac15(Az8)2.
Reviewer 4 Report
The manuscript is well addressed.
Minor comments:
Fig 2C should be quantified and P-Value should be indicated
Fig 6C, D and G addition of the Mice survival graph will be more relevant to the clinical point of view.
Please comment on the toxicity of the drugs. Has it been tested for ?
